# Challenges for Triple Negative Breast Cancer Treatment: Defeating Heterogeneity and Cancer Stemness

**DOI:** 10.3390/cancers14174280

**Published:** 2022-09-01

**Authors:** Rinad Mahmoud, Paloma Ordóñez-Morán, Cinzia Allegrucci

**Affiliations:** 1Centre for Cancer Sciences, Biodiscovery Institute, University of Nottingham, Nottingham NG7 2RD, UK; 2Translational Medical Sciences Unit, School of Medicine, University of Nottingham, Nottingham NG7 2RD, UK; 3Nottingham Breast Cancer Research Centre, Biodiscovery Institute, University of Nottingham, Nottingham NG7 2RD, UK; 4SVMS, University of Nottingham, Sutton Bonington Campus, Loughborough LE12 5RD, UK

**Keywords:** triple negative breast cancer, stemness, differentiation, persister cells, phenotype, drug resistance, tumour heterogeneity

## Abstract

**Simple Summary:**

Triple Negative Breast Cancer represents a cancer type with an unmet clinical need. This type of breast cancer presents the worse clinical outcome due to its aggressiveness, high heterogeneity, and absence of therapeutic targets. Chemotherapy is still the standard of care for this type of cancer, but many patients develop resistance to treatment and metastatic disease. In this review, we highlight the existing challenges for effective treatment of triple negative breast cancer. We discuss the importance of the stratification into different molecular subtypes and the identification of resistant cells within tumours that is needed to guide future strategies for effective and precise therapies.

**Abstract:**

The Triple Negative Breast Cancer (TNBC) subtype is known to have a more aggressive clinical course compared to other breast cancer subtypes. Targeted therapies for this type of breast cancer are limited and patients are mostly treated with conventional chemo- and radio-therapies which are not specific and do not target resistant cells. Therefore, one of the major clinical challenges is to find compounds that target the drug-resistant cell populations which are responsible for reforming secondary tumours. The molecular profiling of the different TNBC subtypes holds a promise for better defining these resistant cells specific to each tumour. To this end, a better understanding of TNBC heterogeneity and cancer stemness is required, and extensive genomic analysis can help to understand the disease complexity and distinguish new molecular drivers that can be targeted in the clinics. The use of persister cancer cell-targeting therapies combined with other therapies may provide a big advance to improve TNBC patients’ survival.

## 1. Heterogeneity in TNBC

Breast Cancer (BC) comprises a group of diverse breast diseases in terms of molecular characteristics, clinical presentation, and therapeutic response. Distinct biological and clinical BC subtypes have been identified using gene expression profiling, molecular pathology, and histopathology [1]. The molecular subtyping facilitates BC patients’ stratification and assists in treatment tailoring to improve patients’ response to therapy [2]. The Triple Negative BC (TNBC) subtype includes BC tumours that have a negative expression status of the Oestrogen Receptor (ER), Progesterone Receptor (PR), and Human Epidermal Growth Factor 2 (HER2) (ER−, PR−, HER2−) [2,3]. According to epidemiological studies, TNBC corresponds to 10–20% of all BC cases and is commonly diagnosed in younger patients with an onset age of 35 years or less [4]. Patients usually present at later stages of the disease [5], and TNBC tumours are known to have a more aggressive clinical course compared to other subtypes [6]. TNBC tumours are usually of a basal histopathological subtype [1,7] and are diagnosed at high pathological grade and tumour stage (stages III/IV) [5,7]. At the time of diagnosis, these tumours more frequently show lymph node involvement [7,8] and the lymph node metastatic state has been identified as an independent TNBC prognostic factor [7]. Irrespective of their high initial clinical response rates to neoadjuvant therapy, TNBC patients still show a worse prognosis with a higher risk of distant metastasis relative to other BC patients [5,9,10] and a short Disease-Free Survival (DFS) when presenting at a younger age [7]. A study from the University of Toronto estimated that TNBC distant recurrence after treatment peaked at approximately three years’ time and showed that TNBC had a mortality rate of about 40% within 5 years of diagnosis [11]. Conventional therapies are in many cases not effective, leading to tumour recurrence due to long-term residual TNBC cancer cells in primary tumours and/or metastatic lesions [12,13]. Therefore, there is an urgent need to develop effective treatments for this aggressive BC subtype [14].

Currently, most TNBC molecular subtyping studies are based on tumours’ gene expression (GE) clustering [2]. TNBC tumours have been categorised into four to six distinct molecular subtypes based on the expression of characteristic GE profiles [14,15,16]. In 2011, Lehmann et al. classified TNBC tumours into six molecular subtypes using GE cluster analysis in a large number of TNBC samples (14 patients’ publicly available RNA profiling datasets and confirmed their results by using seven other available datasets, *n* = 386). Based on gene ontologies and differential GE, the analysis revealed seven tumour clusters, with six stable clusters each displaying a distinctive gene signature, namely: the mesenchymal (M), mesenchymal stem-like (MSL), basal like-1 (BL-1) and -2, (BL-2), luminal androgen receptor (LAR), immunomodulatory (IM) subtype, and one additional unstable (UNS) tumour cluster that expresses genes that are found across the other six clusters [14]. The study showed that with an adequate patient sample size, TNBC GE analysis can determine distinct subtypes and reveal molecular targets, providing predictive biomarkers to help stratify patients for tailored treatments and, consequently, help improve patient response to therapy with the design of appropriate clinical trials [14]. Lehmann et al. also used the TNBC patient tumours derived GE signature to reveal TNBC cell lines for each of these subtypes which represent clinically relevant models for the functional testing of novel targeted agents. The molecular stratification within the six molecular subtypes described by Lehmann et al. also identified a subgroup with aberrant PTEN expression, five specific microRNA aberrations, high MYC expression, TP53 mutation, RB1 loss, and WNT signalling. These tumours were linked with poor clinical outcomes. Indeed, poor prognosis and high hazard ratios were found associated with PTEN-low/miRNA-low high RhoA signalling in the BL-1 tumours, AKT1 copy gain/high mRNA expression in the BL-2, and high programmed cell death 1 (PD1) expression in IM [17].

Further studies have classified TNBC into different subtypes. In 2015, Burstein et al. conducted RNA and DNA profiling analyses on 198 TNBC tumours from Baylor College of Medicine (Houston, TX, USA) and confirmed their results by using seven publicly accessible TNBC datasets [15]. Their analysis identified four distinct TNBC subtypes including mesenchymal (MES or cluster 1), luminal androgen receptor (LAR or cluster 2), basal-like immunosuppressed (BLIS or cluster 3), and basal-like immune-activated (BLIA or cluster 4). Each of these stable TNBC subtypes was characterised by the expression of distinct molecular profiles with distinct prognoses: BLIA tumours with the best outcome and BLIS with the worst prognosis. These authors compared their work to Lehmann’s results and found comparable results. They showed that cluster 1 included LAR tumours, clusters 2 and 3 included MSL tumours and some claudin-low M tumours, cluster 3 and cluster 4 included both BL-1 and BL-2 tumours without being separated as different subtypes, and, finally, IM tumours could be found in clusters 2 and 4 [15]. Interestingly, both labs found stromal, immune, and basal gene clusters.

A subsequent study performed hierarchical clustering analyses including 2188 genes from Lehmann’s study and its own TNBC dataset allowing the identification of four main GE clusters: luminal, immune, basal epithelial, and stromal signature clusters [18]. Similar hierarchical clustering was done with the 2188 genes and a dataset containing BC samples, xenografts, normal breast samples, and breast cancer cell lines. This test revealed that any of the cell lines or xenografts expressed the high expressed genes of the stromal/MSL or IM gene signature. This was the first experimental evidence showing that the tumour samples belonging to the IM and MSL signature could be contaminated with cells of the tumour microenvironment, such as fibroblasts or immune cells [18]. Indeed, tumour infiltrating lymphocytes (TILs) greatly contribute to gene expression profiles, and correlations to this signature were proposed to be a descriptor of the immune state of TNBC tumours rather than an independent subtype [19]. Immune infiltration also positively affects tumour prognosis [20,21,22], supporting the predictive value of TILs-related gene expression for better relapse-free survival (RFS) irrespective of the TNBC subtype [21,23]. Later on, Lehmann et al. confirmed these findings and refined the molecular subtypes into four different ones: BL-1, BL-2, M, and LAR, thus excluding the IM and MSL as intrinsic subtypes (Figure 1) [24].

TNBC subtyping and classification by Lehmann et al. was revisited in 2021 [19]. Further analysis of TNBC subtypes helped to identify novel driver signalling pathways in each subtype, highlighting targets for TNBC therapy. The study used DNA copy number genomic and epigenomic data analyses, as well as scRNA-sequencing [14]. Using unsupervised k-means consensus clustering of the TCGA TNBC GE data (*n* = 192), five distinct tumour clusters were identified with some showing a degree of overlap with the previously identified subtypes. Cluster 1 was primarily the M-subtype, cluster 2 contained a combination of M- and BL-1 subtypes, cluster 3 was a mixture of BL1- and immunomodulatory (IM)-subtypes, cluster 4 was predominantly the BL-2 subtype, and cluster 5 contained the LAR subtype. Interestingly, invasive ductal carcinoma was the most common histology across molecular subtypes [19]. Nevertheless, some special histological subtypes were significantly enriched in individual subtypes, thus indicating their diversity and prognostic variability [19]. These pathological types were previously correlated with poor disease outcomes, with histological features of prognostic and predictive value in TNBC [25,26].

### 1.1. The Basal-like (BL) Subtypes

Cytokeratin’s expression can be used to classify BCs as BL or luminal [27]. TNBC subtypes display differential expression of luminal cytokeratins (KRT7, 8, 18, and 19) and BL cytokeratins (KRT5, 6A, 6B, 14, 16, 17, 23, and 81). The BL subtypes (BL-1 and BL-2) express higher levels of basal cytokeratin compared to other subtypes [14]. Despite the similarity in the basal markers of GE profiles, the two BL subtypes are categorised under different histopathological subtypes with BL-1 tumours identifying medullary carcinomas, and BL-2 metaplastic carcinomas [19]. Furthermore, BL-1 tumours were noted to be closely related to the bi-potent L1.2 luminal progenitors, whereas BL-2 tumours have a myoepithelial cell origin [19]. This is in line with prior BRCA-mutated mouse model studies which proposed luminal cells as the cell of origin of basal-type BC. BL subtypes differ in that the BL-2 subtype expresses unique gene ontologies comprising glycolysis and gluconeogenesis signalling as well as growth factor signalling (NGF, EGF, MET, IGF1R) and Wnt/β-catenin pathways. In addition, the BL-2 subtype distinctively expresses high levels of growth factor receptor genes tyrosine-protein kinase Met (MET), epidermal growth factor receptor (EGFR) and EPHA2, membrane metallo-endopeptidase (MME/CD10), and TP63, features that are suggestive of a basal/myoepithelial origin [14].

BL are predominantly characterised by genomic instability, with a number of subtype-specific cell lines showing nearly 2-fold chromosome rearrangements compared to all other subtypes [28]. In addition, the BL subtypes were found to express high levels of proliferation and DNA damage response genes, suggesting that these tumours could benefit from therapies targeting highly proliferative cells such as DNA-damaging and anti-mitotic agents. Patients with BL tumours undergoing radiation-based and taxane-based treatment have about a 4-fold higher pathologic Complete Response (pCR) relative to patients’ tumours with M or LAR subtype characteristics [29,30]. Thus, specific markers to identify DNA damage response signalling defects and proliferation biomarkers could help the stratification of patients for selective and tailored BL cancer treatments [14].

### 1.2. The Mesenchymal (M) Subtype

The M subtype characteristically expresses genes involved in epithelial-to-mesenchymal transition (EMT) [14,31]. M tumours are frequently of malignant phyllodes histology, which have a worse prognosis compared to the other subtypes [14]. These tumours also show high dependencies on adhesion/motility and growth factor genes [19]. Indeed, the M subtype has increased active cell migration-associated signalling pathways regulated by actin protein, a high expression of extracellular matrix-receptor interacting pathways, and differentiation pathways, including the Wnt/β-catenin pathway, the transforming growth factor-β (TGF-β) signalling pathway, and the anaplastic lymphoma kinase pathway [2,14]. Therefore, it has been suggested that M subtype patients could be treated with EMT-targeting agents [2]. In addition, the M subtype highly expresses stemness-related genes and high levels of genes that are involved in developmental processes [14], thus showing dependency on retinoic acid receptors [19]. Recently, it has been proven that drug combinations comprising retinoic acid derivatives and γ-secretase inhibitors (Notch pathway inhibitors) show a synergistic effect on TNBC in vitro and in vivo models [19,32]. NOTCH1/2/3 mutations are frequently seen in the M and BL-1 subtypes. Mutations involving the NOTCH PEST domain are known to be oncogenic and are key to TNBC sensitivity to γ-secretase inhibitors [33]. Therefore, targeting NOTCH could be an effective therapeutic approach for M and BL1 tumours showing NOTCH mutations [19], especially when combined with retinoic acid derivatives [32]. The M subtype also shows a negative immune signature, with the low expression of genes involved in antigen processing and presentation, interferon-gamma response, and T cell signalling. This indicates the ability of this subtype to evade the immune response and limitedly respond to chemotherapy compared to other immune-infiltrated TNBC subtypes [19].

### 1.3. The Luminal Androgen Receptor (LAR) Subtype

The LAR subtype is classified according to the AR gene signature [34]. LAR subtype tumours highly express luminal cytokeratins. These tumours also express other luminal markers (XBP1 and FOXA1) but lack the expression of basal cytokeratins [14]. RNA-seq clustering analysis determined that LAR tumours are consistent with the differentiated luminal state and their gene signature is closely related to the L2 hormone-responsive cells. This explains the dependency of LAR cells on hormone signalling [19]. LAR subtype tumours were noted to be diagnosed at an older age compared to other subtypes and are more frequently of the invasive lobular histopathological subtype [19].

## 2. TNBC and Drug-Resistant Cells

### 2.1. Stem Cells and Cellular Origin of TNBC

The cellular origin of TNBC is still debated [35]. It is unclear whether the different subtypes of TNBC originate from mammary stem cells or progenitor cells. Stem cells (SCs) are unspecialised cells that have the ability to self-renew and are able to differentiate into the different cell types comprising the body’s tissue [35,36]. In normal human tissue, SCs give rise to new cells to preserve healthy organs. Similarly, in cancer, the SCs maintain the persistence of malignant tumours by producing more cancer cells [37]. The process of tumour initiation can be driven by the transformation of tissue-resident SCs [38]. This cellular transformation can occur during tissue regeneration. Alternatively, could be initiated and/or accelerated in response to metabolic dysregulation, toxins, infections, or radiation leading to genomic mutations [39]. In 2003, Al-Hajj et al. identified a cellular subpopulation in breast tissue with the ability to initiate tumours. These cells were called “tumorigenic cancer cells” [40]. By implanting human BC cells in immunocompromised mice, it was noted that the ability to form new tumours was constrained to a subgroup of BC cells that uniquely expressed the surface makers CD44^+^/CD24^−^ [40]. As few as 100 cells with this phenotype had the ability to form tumours in mice, comprising phenotypically diverse mixed populations of nontumorigenic cells present in the initial tumour as well as additional CD44^+^/CD24^−^ tumorigenic cells [40]. Since then, the tumorigenic ability of isolated CD44^+^/CD24^−^ cells has been repeatedly confirmed in primary tissues [41,42], and human BC cell lines [43,44]. These cells were described in later work as the “human mammary stem/progenitor cells” [42].

A similar population was identified in 2006 [45]. Shackleton et al. discovered rare mammary stem cells (MaSCs) within the mouse breast tissue that do not express the endothelial marker CD31, nor the hematopoietic markers CD45 and TER1. Cells, expressing high levels of CD29 (β1 integrin) and the epithelial marker CD24, can reconstitute alveolar-like structures that produce milk protein. These cells are also able to generate neo-breast tissue and maintain a stable pool of tissue-resident stem cell progenitors. This self-renewal and multipotency capacity is also a property of human BC SCs (CSCs), which can be demonstrated in BC subtypes sharing gene ontologies similar to MaSCs [45].

Independent of breast tumour molecular subtypes, breast CSCs exist in distinctive mesenchymal-like and epithelial-like states based on their expression of CD44, CD24, and Aldehyde Dehydrogenase 1 (ALDH) markers [46]. RNA-seq analysis and immunofluorescent staining of BC samples showed that the mesenchymal-like CSCs mainly express CD44^+^/CD24^−^ and are primarily inactive/quiescent and localised at the edges of tumours. Contrarily, the epithelial-like CSCs are proliferative, centrally located, and highly express ALDH genes [46]. Breast CSCs display high plasticity that allows them to transition between the epithelial-like and mesenchymal-like states [46]. Interestingly, an abundance of CD44^+^/CD24^−^ cells has been reported in TNBC tumours compared with luminal and HER2 subtypes [47]. It has been hypothesised that TNBC subtype growth originates from a CSCs population or tumour-initiating cells harbouring oncogenic gene mutations that are critical for tumour growth and response to therapy [46,48,49]. Indeed, the enrichment of TNBC tumours with CD44^+^/CD24^−^ cells confers them to a higher proliferation, migration, invasion, and tumorigenic capacity [50]. CSCs play an important role in the TNBC’s aggressive behaviour. Mouse studies showed that only 2% of these tumour-initiating cells can form secondary tumours [50]. Of note, this aggressive cellular population is mainly seen in the BL2 and MSL TNBC subtypes [14,18]. GE and gene ontology analysis showed enrichment in EMT and stemness-related pathways in these subtypes, including FGFR, mTOR, TGF-β, Rac1/Rho, Wnt/β-catenin, PDGFR, and VEGF signalling [14]. These pathways are commonly seen in normal mammary tissue within CD44^+^/CD24^−^ cells [51]. Different studies have identified the importance of SC markers and their prognostic role in TNBC [52]. Indeed, CSC markers have been strongly associated with advanced tumour stage, tumour size, higher tumour grade, metastasis, and lymphatic involvement in TNBC patients [52,53].

### 2.2. Persister Cells in TNBC

Recently, new studies have been focussing on a discrete population of cells within tumours, known as “persistent cancer cells”. These are usually undetected cells that survive cancer therapy and are considered a major cause of treatment failure [54]. Acquired resistance is seen in many pathologies including infectious diseases and malignancies, often due to long treatments that allow the selection of these persistent/resistant cells [55]. It was recently demonstrated by Ramirez et al. that drug tolerance is a phase between therapeutic sensitivity and resistance from which resistant/persistent clones can emerge [56]. These cells can survive and develop progressive drug tolerance, thus gaining the ability to expand during treatment and correlating with high cancer recurrence [55,57] (Figure 2). Interestingly, it was suggested that the terms “quiescent”, “dormant”, “tolerant”, and “persister” cells in cancer all describe one discrete tumour cell population [55]. In some cases, these cells display similarities with the molecular profile of SCs [58] and therefore are described as “cancer stem-like cells” [55]. Links between dormancy and stemness properties have been already established in numerous cancer types, including BC [59,60,61].

Persister cells are highly flexible in their energy consumption and adaptation to their microenvironment [54]. They have slow proliferation rates due to their quiescent properties but have the capability to re-enter the cell cycle, which enables them to proliferate, giving rise to tumour relapse [62]. Reduction in the proliferation rate provides a selective advantage to resist treatment, thus triggering enrichment of dormant cells with a stem-like phenotype [63]. Additional properties by which persister cells resist include their ability to hijack their environment by creating an immune-tolerant niche [64,65]. The mechanisms involved in the selection process are still unclear, but it is becoming increasingly evident that resistance is associated with the heterogeneity of cancer cells and that multiple mechanisms underlie the emergence of drug-resistant subpopulations [55]. Mechanisms that trigger their persistence offer highly sought-after therapeutic targets, including epigenetic, transcriptional, and translational regulatory processes, as well as complex cell-cell interactions [54].

Resistance to conventional therapies is commonly seen in TNBC tumours [12,13]. A residual resistant genotype that is adaptively selected by chemotherapy has been suggested to be responsible for failure of treatment. This was proved by a single cell sequencing analysis obtained from TNBC patients showing adaptive selection by neo-adjuvant chemotherapy treatment [66] and leading to high treatment failure in TNBC patients. Furthermore, numerous pathways are known to regulate TNBC CSCs survival [66,67], including Hedgehog [68], Wnt/β-catenin [65,69], JAK/STAT [70,71], and HIPPO pathways [72]. Despite significant progress being made in understanding the mechanism behind persistence, there is still an urgent need for successful clinical targeting of these specific cancer cells.

An interesting mechanism by which TNBC tumours resist therapy is therapy-induced senescence (TIS) [73]. Senescence, or cellular growth arrest, is a cellular fate initially discovered in the context of cultured cells growth arrest and is now being recognised as an important mediator of numerous physiological and pathological processes [74]. It is believed that oncogene-induced senescence (OIS) is one of the contributing factors to TIS in numerous types of cancer, including mammary tumours [75]. Indeed, senescence can promote cancer stemness and tumour aggressiveness [76], with the stem-like state identified as the mediator for the development of drug-resistant aggressive clones within these tumours [73]. A study using matched pair gene expression analysis of 17 primary BC biopsies pre- and post-neoadjuvant chemotherapy revealed enrichment of TGF-β signatures, a cytokine that has been associated with breast CSCs in treated samples [77]. This unique gene signature was also noted to be similarly altered in the human TNBC cell line SUM159 following treatment with paclitaxel, thus suggesting an enrichment of the CSCs population with an upregulation in genes involved in the TGF-β pathway following chemotherapy treatment [77]. It is believed that TGF-β is key for TIS due to its senescence-promoting autocrine/paracrine role in aging/aging-related pathologies [77]. Numerous potential TGF-β inhibitors are currently being tested in clinical trials as possible novel therapies to improve TNBC patient prognosis [78]. These pieces of evidence suggest that targeting senescence could be further exploited in TNBC cancer therapy [73].

## 3. Novel Therapeutic Approaches for TNBC

Despite the advancements in the discovery of new therapies, TNBC patients’ treatment remains extremely challenging [13]. This is due to the high heterogeneity of the disease and the lack of receptor expression that is targeted by available therapies [15]. TNBC tumours show no response to endocrine (hormonal) therapy or HER-2 targeting agents, hence chemotherapy remains the main systemic course of treatment [12]. Chemotherapy can be given in the neoadjuvant and/or adjuvant setting and there are no major differences in patient’s survival probability between these two types of treatment [79]. However, neoadjuvant chemotherapy is currently regarded as the standard therapeutic approach for high-risk TNBC as it helps reducing primary and metastatic tumour burden prior to surgical resection [12,79,80], as well as assessing tumour response and the potential need for adjuvant treatments [13]. Recently, a key role of immune-checkpoint inhibitors has been defined in cancer treatment [13,81,82]. BC immune-gram is suggested to be a potential application that assesses the tumour microenvironment and helps implement precision immunotherapy. Currently, ongoing trials are testing different combinations that will improve immunotherapy efficacy [13].

Combination therapy is also currently proving effective to increase the effect of innate immune responses against TNBC. Indeed, a viral mimicry response can be induced when using epigenetic inhibitors (histone methyltransferase EZH2 and PRMT1 inhibitors) that induce the expression of transposable element-derived double-stranded RNA. Such a response is able to activate an interferon response resulting in a potent antitumour effect [83]. Furthermore, a study has recently linked H3K4me3 (trimethylation of histone H3 at lysine 4) and H3K27me3 (trimethylation of histone H3 at lysine 27) to the persister cell population expression program in TNBC tumours. H3K27me3 has been identified as a key activator of the persisters transcription program and its suppression reduces chemotherapeutic tolerance in TNBC in vivo models [84]. Current therapies available for TNBC subtypes are described below and summarised in Figure 3.

### 3.1. BL Subtypes

BL1 tumours show an increased number of mutations (average 2.1 mut/Mb and 2.3 mut/Mb, respectively) compared to tumours from other subtypes. However, despite this high mutational load, BL1 tumours are associated with better survival, reflecting their high response to standard chemotherapy [19]. BL1 cell line models were proven to be highly sensitive to the cell cycle inhibitors PHA-793887 (CDK2/5 inhibitor) and ZM447439 (AURKA/B inhibitor) [13]. Cells were also noted to be sensitive to DNA repair pathway suppressors including NU7441 (DNAPK inhibitor) and KU-559333 (ATM inhibitor) [13]. On the other hand, BL2 subtype tumours exhibit a lower genomic copy number complexity and mutational load. The BL2 proteomic and phosphoproteomic data demonstrate an intact G1/S checkpoint. These tumours express high CDK6 protein levels, and their cell lines were noted to be highly sensitive to CDK6 knockdown, indicating that CDK4/6 inhibitors are good candidates for treatment [19]. Furthermore, the BL2 subtype exhibits a distinctive genetic dependency on developmental pathway mutations (KRAS and activating MAPK pathways) and expression of developmental genes (WNT3, JAG1, NODAL, BMPR1A, and RSPO2). This explains the response of the BL2 cell lines to the DNA repair targeting agents CP466722 and Olaparib, and their unique sensitivity to the DNA alkylating agents temozolomide, carboplatin and cyclophosphamide [19]. Furthermore, BL2 TNBC cell lines were noted to be responsive to a set of MAPK pathway inhibitors including PD0325901, trametinib, refametinib, CI-1040, and selumetinib [19].

### 3.2. M Subtype

A decreased 5-year Distant-Metastasis-Free Survival (DMFS) was noted in patients with M tumours and is associated with the enrichment in EMT and motility-related pathways found in this tumour subtype [14]. Moreover, it was previously shown that the EMT mediating gene Src has a prominent role in highly invasive cancer cells that have undergone EMT [87] and cells of the M subtype are sensitive to dasatinib, an Src kinase family inhibitor that is a known potent suppressor of TNBC stem cells [14,88]. Furthermore, EMT can be regulated by the Wnt/β-catenin pathway (APC, CTNNB1, and WISP3) mediating tumour cell invasion [89]. Recently, TNBC tumours were noted to be characterised by a significant differential expression of the Wnt pathway components [90] and this signature is linked to poor patients outcome [69,85]. Therefore, Wnt pathway dysregulation could be used as a therapeutic target [85,90]. Recently, pre-clinical studies and clinical trials demonstrated that concurrent inhibition of the Wnt signalling together with chemotherapy and/or targeted therapies administration has a synergistic effect in TNBC treatment [69]. Therefore, drugs targeting this pathway could be of great value at least for the treatment of M subtype tumours [14]. In addition, M subtype cells are uniquely sensitive to other kinase inhibitors, including midostaurin (targeting FLT3), BX796 (targeting PDK1), SL0101 (targeting RSK), and ponatinib (targeting RTK) [19]. Inhibition of kinase signalling is also effective in reducing tumour growth in patient-derived xenografts, with inhibition of TGFβ, p38/JNK, Rac, and RTK proving to be the most effective [19]. Finally, M tumours are sensitive to retinoic acid, suggesting a strong epigenetic dependence of this tumour subtype. This is also demonstrated by the strong effect of the histone methyl transferase EZH2 inhibitors (tazemetostat and CPI-1205) in reducing tumour growth and inducing epigenetically-regulated immune responses [19].

### 3.3. LAR Subtype

Retrospective studies showed a lower pathological grade and complete response rates to neoadjuvant chemotherapy for AR-expressing tumours [91]. This was explained by genomic studies where it was noted that LAR tumours carry lower mutational burdens and are more genetically stable. Protein analysis showed that LAR tumours display low activation of the cell cycle [19]. Furthermore, LAR tumours and cell line models display high dependencies on AR, FOXA1, ERBB2, and AKT protein and phospho-protein signalling, as well as frequent PIK3CA and ERBB2 mutations [19]. AR antagonist in vitro testing in five different LAR cell line models showed high sensitivity to 17-dimethylaminoethylamino-17-demethoxygeldanamycin (17-DMAG) and bicalutamide treatment, thus indicating that AR targeting therapies may be effective against LAR tumours [14]. Currently, the next AR antagonist generation enzalutamide is being tested on AR-positive (AR+) tumours as a combined therapy with paclitaxel (NCT02689427) [92]. LAR cell lines were also noted to be sensitive to PI3K inhibition as PIK3CA activating mutations make them sensitive to the PI3K inhibitor NVP-BEZ235 [31]. Therefore, simultaneous targeting of PI3K/mTOR and AR signalling may be of clinical value for LAR tumours [14]. A recent clinical trial using enzalutamide and the PI3K inhibitor, taselisib, as a combined therapy showed an enhanced clinical outcome rate in metastatic AR+ TNBC patients [19,86]. Moreover, despite the absence of amplifications, the existence of ERBB2 mutations and increased protein levels, indicate that ERBB2 inhibitors may be an alternative therapeutic approach for LAR subtype tumours [19]. Despite demonstrating lower copy number complexity and mutational levels relative to other subtypes, proteomics and phosphoproteomics data suggest an intact G1/S checkpoint. However, LAR cell lines show genetic dependence on CCND1 and CDK4. Importantly, the CDK4/6 inhibitor Ribociclib is currently being tested in combination with bicalutamide in AR+ TNBC patients (NCT03090165) [19,86].

### 3.4. Targeting the CSCs and/or Persistent Cell Population in TNBC

CSCs have been highlighted as key drivers of TNBC aggressiveness [50]. Therefore, the identification and regulation of CSCs could be a promising therapeutic strategy for TNBC in the future [48]. Fasting-mimicking diet (FMD) is a new strategy developed to reduce CSCs cells in the early TNBC stages [93]. Indeed, FMD can reduce glucose-dependent PKA signalling in TNBC SCs by inducing hypoglucemia and reducing CSCs and tumour progression [93]. Contrarily, in differentiated cancer cells, FMD stimulates starvation escape pathways, including mTOR, PI3K/AKT, and CDK4/6, thus providing therapeutic targets that can lead to tumour regression with low toxicity [93]. This was further supported by evidence that low basal glucose levels are accompanied by high survival rates in metastatic TNBC patients [93,94].

Modulating the tumour stromal microenvironment is another approach by which TNBC resistance could be targeted. The extracellular matrix (ECM) is a known regulator of the hallmarks of cancer [20,95]. The degree of ECM stiffness is well known to greatly impact the intrinsic cell behaviour by influencing growth factor signalling. ECM stiffness also modulates drug responses through controlling blood vessel transport, as ECM is known to induce angiogenesis, hypoxia, and compromise anti-tumour immunity [96]. Primary TNBCs are surrounded by a rigid stromal microenvironment whereas chemotherapy-resistant residual tumours populate a softer niche, which contributes to drug resistance [97]. This effect is linked to NF-κB activity which mediates suppression of the pro-apoptotic protein JNK, thus suggesting that targeting the biophysical properties of ECM by NF-κB inhibition could enhance patients’ therapeutic response [97].

Recently, new treatments that target chemotherapy-resistant TNBC CSCs have been introduced to enhance chemosensitivity and improve outcomes. A novel multi-kinase (CK2/TNIK/DYRK1) inhibitor 108600 that targets the CSC population represses the growth, colony, and mammosphere formation by inducing G2M arrest and apoptosis [98]. The compound also shows excellent results in in vivo models, as 108600 treatment overcomes chemotherapy resistance in mice bearing TNBC tumours. Strong evidence for clinical translation of this agent into clinical trials was provided by its ability to suppress the growth of pre-established metastases in in vivo models [98]. Furthermore, a novel combination therapy comprising an RAF/MEK inhibitor CH5126766 (VS-6766) and eribulin has been demonstrated to potently inhibit TNBC cell line growth by inducing apoptosis and simultaneously suppressing the expression of the programmed cell death ligand 1 (PD-L1), with significant reduction in tumour growth in vivo [99] (Figure 4).

## 4. Conclusions

A great advance in the management of TNBC has been achieved in the last decade, especially by using the molecular characterisation of tumours to guide strategies for the development of specific therapies. The growing perception of the importance of tumour heterogeneity, the degree of stemness and differentiation, and the development of persister cancer resisting therapies are opening new avenues to targeted therapies for TNBC treatment. The molecular profiling of TNBC subtypes has revealed therapeutic vulnerabilities that can be pursued to drive the precise treatment of patients that can be stratified according to the expression of subtype-specific gene signatures and mutations. However, the nature of CSCs in TNBC still remains unclear as stemness and self-renewal are complex processes established and maintained by dynamic molecular networks and the tumour microenvironment. Identifying new contributors to TNBC stemness and drug resistance may therefore assist in resolving this complexity and help identify new effective TNBC therapeutic strategies.

## Figures and Tables

**Figure 1 cancers-14-04280-f001:**
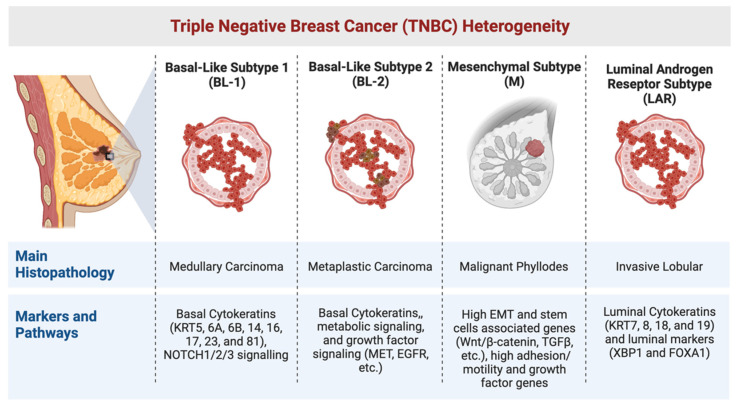
TNBC heterogeneity, showing TNBC subtype’s main histopathology, markers, and signalling pathways according to Lehmann et al. 2021 [19]. Created with BioRender.com, agreement number MS248V9M9Z.

**Figure 2 cancers-14-04280-f002:**
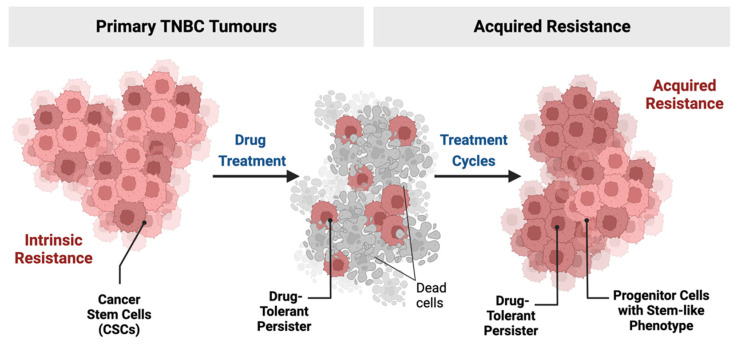
TNBC tumours exhibit intrinsic resistance to therapy due to the presence of cancer stem cells as well as acquired resistance after treatment cycles. Created with BioRender.com, agreement number QB248V9UMD.

**Figure 3 cancers-14-04280-f003:**
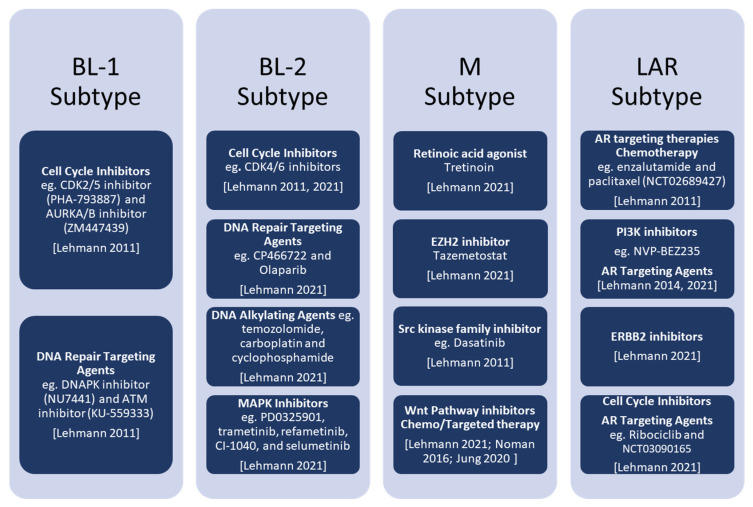
Current therapeutics available for TNBC subtypes [14,19,31,68,85,86].

**Figure 4 cancers-14-04280-f004:**
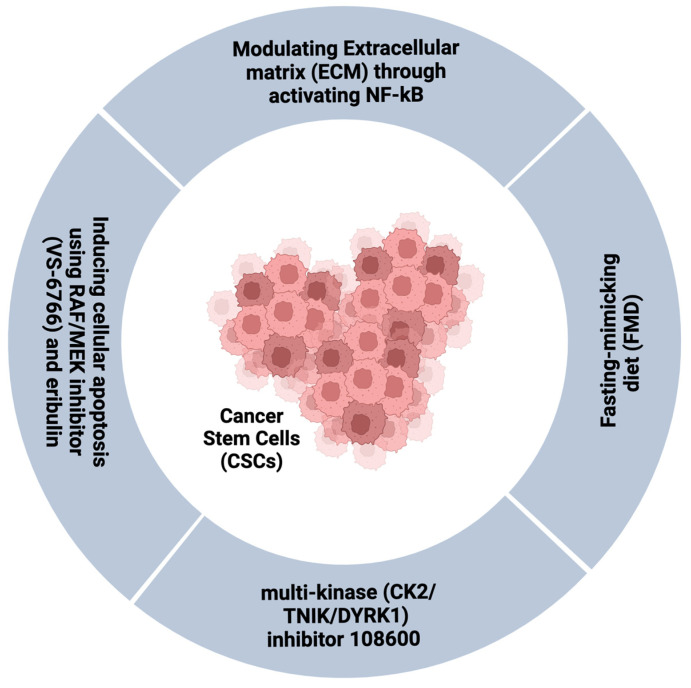
Therapeutic strategies to target CSCs in TNBC. Created with BioRender.com, agreement number IQ248V9PR7.

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
