# Peer review of "Challenges for Triple Negative Breast Cancer Treatment: Defeating Heterogeneity and Cancer Stemness"

_cancers, 2022, doi:10.3390/cancers14174280_

Round 1

Reviewer 1 Report

Overall, this is a comprehensive, well-researched review article.  It describes the heterogeneity of TNBC, basis of TNBC subtype classification, and how the molecular characteristics could impact response to therapeutic agents. It also reviews advances in drug resistance, ‘persister’ cancer cells, and cancer stem cells. However, there are some important concerns that need to be addressed:

To start with, the title of the article is confusing and misleading. The authors have not justified the phrase “Defeating heterogeneity and cancer stemness” and how this makes TNBC vulnerable to therapy. Instead of this wordy title, this reviewer suggests a simpler and straightforward tile like: “ Tumor heterogeneity, stemness and therapeutic resistance in Triple negative breast cancer” that may be more appropriate given the content of this review article.

The authors describe the TNBC classification put forth by Lehman et al (2014), Burstein at al (2014), and revised classification by Lehman et al (2016 and 2021).  Although the authors mention that in 2021 Lehman et al revisited their original classification (in 2014), they do not describe the fact that Lehman et et al.,’s original classification of TNBC to six molecular subtypes (in 2014) was refined by them (in 2016 and in 2021) from six molecular subtypes into four tumor-specific subtypes (BL1, BL2, M and LAR). As the immunomodulatory (IM) and mesenchymal stem-like (MSL) subtypes in the were contributed from infiltrating lymphocytes and tumor-associated stromal cells, respectively, IM and MSL are no longer considered as independent subtypes. The authors cite Lehman et al 2021 paper, but not the 2016 paper (manuscript line 82). In spite of the revised classification into BL1, BL2, M and LAR groups, the authors still describe in detail six subtypes (BL1, BL2, IM, M, MSL, and LAR). The entire section should be revised in the context of the Lehman et al and Burstein et al classification. The authors should provide a critical comparison between the Lehman et al and Burstein et al classification.   Figure 1 should be modified accordingly as well.

Furthermore, the authors erroneously state that Perou et al in 2019 performed molecular stratification ….. (manuscript line 74). That particular work referred to was performed by Dong-Yu Wang et al (2019), not by Perou’s team.  

Section 4. Conclusions: (manuscript lines 461 to 469). This section needs to be revised to make it more consistent with theme and descriptions in the review article. 

The sentence “This was previously anticipated to be the biological principles determining poor outcomes in this pathology, where histological features has a known prognostic and predictive value in TNBC [20],[21]” (manuscript lines 95-97) is vague (besides grammatical error) and it not clear what the authors are trying to convey.

There are several grammatical and sentence errors scattered all over the manuscript. This reviewer suggests they be corrected in consultation with someone with mastery in English language. Following sentences are just a few examples:

“The cellular origin of TNBC is still debated and it is unclear to make it more appropriate  if any of the different subtypes of TNBC are the origin from mammary stem cells or progenitor cells”.

“The reduction in the proliferation rate provides a selective advantage to resist to drug pressure which would trigger the enrichment of dormant cells with a stem-like phenotype, as described previously by Zhou et al. in other tumour types” (manuscript sentence 273).

“….the next AR antagonist generation enzalutamide is being tested on AR positive 398 (AR+) tumours as a combined therapy with paclitaxel (NCT02689427)” .

“..the next AR antagonist generation enzalutamide is being tested on AR positive 398 (AR+) tumours as a combined therapy with paclitaxel (NCT02689427)”

Reviewer 2 Report

Comments to the Authors:

The review article on “Therapeutic vulnerability of Triple Negative Breast Cancer: defeating heterogeneity and cancer stemness” describes about TNBC heterogeneity, the reason for drug resistance and the therapies to target TNBC. The article is presenting all the aspects to understand disease complexity and distinguish new molecular drivers that can be targeted in the clinics. The review can be modified a little by incorporating few suggestions.

Minor concerns:

• Please provide a table on their large section on “1.Heterogeneity in TNBC”.

• Please undergo a thorough check of the manuscript for typographical and grammatical errors.

Reviewer 3 Report

Reviewer comments:

Comments to the Author

The review by Dr. Mahmoud et al., emphasizes on understanding TNBC heterogeneity with genomic analysis to understand disease complexity and distinguish new molecular drivers that can be targeted in the clinics.

This review article is impressive, and for the most part well written with substantial evidence of literature available. The discussion is also well goes with the results and postulated according to the evidence provided. The references are appropriate and timely.

Minor criticisms

• Please undergo a thorough check of the manuscript for typographical and grammatical errors.

Round 2

Reviewer 1 Report

The authors have effectively addressed the major concerns in the original review.